# The *Sporisorium reilianum* Effector Vag2 Promotes Head Smut Disease via Suppression of Plant Defense Responses

**DOI:** 10.3390/jof8050498

**Published:** 2022-05-11

**Authors:** Yulei Zhao, Nisha Agrawal, Hassan Ghareeb, Mohammad Tanbir Habib, Sascha Dickmeis, Jens Schwachtje, Tim E. Iven, Joachim Kopka, Ivo Feussner, Jan Schirawski

**Affiliations:** 1Department of Molecular Biology of Plant-Microbe Interactions, Albrecht-von-Haller Institute for Plant Sciences, University of Göttingen, Julia-Lermontowa-Weg 3, 37077 Göttingen, Germany; zhaoyulei617@gmail.com (Y.Z.); hassan.ghareeb@kws.com (H.G.); tanbir_ru@yahoo.com (M.T.H.); 2Department of Microbial Genetics, Institute of Applied Microbiology, RWTH Aachen University, Worringer Weg 1, 52074 Aachen, Germany; nisha.agrawal@uni-jena.de (N.A.); sascha.dickmeis@web.de (S.D.); 3Department of Genetics, Matthias Schleiden Institute, Friedrich Schiller University Jena, Philosophenweg 12, 07743 Jena, Germany; 4Max-Planck-Institute for Terrestrial Microbiology, Karl-von-Frisch-Str. 10, 35043 Marburg, Germany; 5Max-Planck-Institute of Molecular Plant Physiology, Potsdam Science Park, Am Mühlenberg 1, 14476 Potsdam, Germany; schwachtje@web.de (J.S.); kopka@mpimp-golm.mpg.de (J.K.); 6Department of Plant Biochemistry, Albrecht von Haller Institute for Plant Sciences, Göttingen Center for Molecular Biosciences (GZMB), University of Göttingen, Justus-von-Liebig-Weg 11, 37077 Göttingen, Germany; tim.e.iven@googlemail.com (T.E.I.); ifeussn@uni-goettingen.de (I.F.); 7Service Unit for Metabolomics and Lipidomics, Göttingen Center for Molecular Biosciences (GZMB), University of Göttingen, Justus-von-Liebig-Weg 11, 37077 Göttingen, Germany

**Keywords:** effector, salicylic acid, smut, *Sporisorium reilianum*, virulence, gene cluster, plant-microbe-interaction, phytopathogenic fungi, metabolic rewiring, plant defense

## Abstract

Genome comparison between the maize pathogens *Ustilago maydis* and *Sporisorium reilianum* revealed a large diversity region (19-1) containing nearly 30 effector gene candidates, whose deletion severely hampers virulence of both fungi. Dissection of the *S. reilianum* gene cluster resulted in the identification of one major contributor to virulence, *virulence-associated gene 2* (*vag2*; *sr10050*). Quantitative reverse-transcriptase polymerase chain reaction (qRT-PCR) experiments revealed high expression of *vag2* during biotrophic growth of *S. reilianum*. Using the yeast secretion trap assay, we confirmed the existence of a functional signal peptide allowing protein secretion via the conventional secretory pathway. We identified the cytoplasmic maize chorismate mutase ZmCM2 by yeast two-hybrid screening as a possible interaction partner of Vag2. Interaction of the two proteins in planta was confirmed by bimolecular fluorescence complementation. qRT-PCR experiments revealed *vag2*-dependent downregulation of salicylic acid (SA)-induced genes, which correlated with higher SA levels in plant tissues colonized by Δ*vag2* deletion strains relative to *S. reilianum* wildtype strains. Metabolite analysis suggested rewiring of pathogen-induced SA biosynthesis by preferential conversion of the SA precursor chorismate into the aromatic amino acid precursor prephenate by ZmCM2 in the presence of Vag2. Possibly, the binding of Vag2 to ZmCM2 inhibits the back reaction of the ZmCM2-catalyzed interconversion of chorismate and prephenate, thus contributing to fungal virulence by lowering the plant SA-induced defenses.

## 1. Introduction

*Sporisorium reilianum* is a smut fungus that causes head smut on maize, leading to yield loss. The disease is characterized by the replacement of male and female inflorescences (tassels and ears) with sori full of black teliospores and/or leaf-like structures [1,2]. The fungal spores can survive for several years in the soil [3]. Under favorable environmental conditions, the teliospores germinate to form haploid sporidia of different mating types [4,5]. Sporidia of different mating types can recognize each other and form conjugation tubes that fuse to form filamentous dikaryotic hyphae that can penetrate the plant epidermis [6,7,8]. The invading hyphae do not disrupt the integrity of the host cells; instead, they grow intracellularly surrounded by the host cell plasma membrane [8] and spread inside the host plant without inducing apparent symptoms until the plant sets flowers.

Like other biotrophic plant pathogens that infect and multiply in living plant tissues, *S. reilianum* needs to overcome physical and chemical barriers as well as a sophisticated immune system potent to fight against invading pathogens [9]. Conserved microbial elicitors called pathogen-associated molecular patterns (PAMPs) or microbe-associated molecular patterns (MAMPs) are recognized through membrane-localized pattern recognition receptors (PRRs) of the plant [10,11]. PAMP recognition leads to basal defense reactions via PAMP-triggered immunity (PTI) that includes Ca^2+^-ion influx across the plasma membrane, accumulation of reactive oxygen species, activation of MAP kinase cascades, protein phosphorylation, and changes in gene expression. As a result, the deposition of callose and plant growth defects can be observed later [12].

To establish a compatible interaction that leads to fungal proliferation inside the host, plant pathogens have evolved small, secreted proteins (effectors) that may help to avoid recognition by PRRs or suppress host PTI. Effectors are molecules that alter host-cell structure and function to either facilitate infection, trigger defense responses, or both [13,14]. Recognition of effectors by plant receptors elicits effector-triggered immunity (ETI) [15]. ETI leads to homo-multimerization of intracellular receptors that function as Ca^2+^-channels and leads to a hypersensitive response resulting in host cell death [16]. In addition, plant hormones play an important role in plant defense [17]. Whereas ethylene and jasmonic acid signaling are important for defense against necrotrophs, defense against biotrophic plant pathogens largely depends on salicylic acid (SA) signaling [18]. Congruously, many plant pathogens have evolved strategies to interfere with SA-mediated defenses, including inactivation of the SA molecule, interference with its biosynthesis, or downstream signaling [19].

In maize, SA-dependent defense signaling is induced by pathogen infection through the activation of papain-like cysteine proteases (PLCPs) in the apoplast. Activated PLCPs lead to the liberation of a short peptide, *Zea mays* immune signaling peptide 1 (Zip1), that induces SA-signaling in the plant [20]. The biotrophic fungal maize pathogen *Ustilago maydis* has been shown to actively repress SA-mediated plant defenses within 24 h after plant inoculation [21]. It was shown that *U. maydis* secretes a chorismate mutase into the plant tissue that interferes with SA biosynthesis by redirecting the SA precursor chorismate into the phenylpropanoid pathway [22]. *U. maydis* also possess a salicylate hydroxylase that can directly degrade SA [23]. While this enzyme does not seem to contribute to virulence, it may be part of an SA-sensing system active during saprotrophic growth that includes the transcriptional regulator Rss1 [24].

In the past few years, considerable progress has been made in smut genome sequencing, which allows the bioinformatic generation of large lists with effector gene candidates [2,25,26,27,28,29,30,31,32,33,34,35]. Whereas the identification of putative effectors is relatively easy, elucidating their role during the infection process is not. So far, only a handful of smut fungal effectors has been functionally characterized. These include the secreted apoplastic effectors Pep1 and Pit2 of *U. maydis* that either block the oxidative burst response by directly inhibiting the activity of maize peroxidases [36,37] or inhibit cysteine proteases in the apoplast at the biotrophic interface between fungus and plant [38,39], respectively, as well as the cytoplasmic effectors Tin2 and See1 that were either shown to be involved in redirecting the biosynthesis of host lignin to anthocyanin to suppress lignification during maize colonization [40] or to stimulate DNA synthesis and tumor formation through interference with the MAPK-triggered phosphorylation of the maize cell cycle regulator SGT1 [41], respectively. In addition, for a much larger number of effectors, an involvement in the smut fungal life cycle has been experimentally confirmed, however, their mechanisms of function still need to be elucidated. These include effectors of *S. reilianum* with a suspected [42] or shown role in virulence [43].

*S. reilianum* and *U. maydis* both possess a very large effector gene cluster containing about 30 genes, most of which carry a predicted protein secretion signal. Previously, the cluster was identified as the largest effector gene cluster in the genome of *U. maydis* (cluster 19A) [25] and as the largest divergence region between the genomes of *S. reilianum* and *U. maydis* (diversity region 19-1) [2]. Deletion of cluster 19A in *U. maydis* led to severely reduced seedling leaf tumor formation in maize [25,44]. Dissection of cluster 19A of *U. maydis* led to the identification of nine effector genes with a measurable contribution to leaf tumor formation. These tumor-inducing genes were named *tin1-1*, *tin1-2*, *tin1-3*, *tin1-4*, *tin1-5*, *tin2*, *tin3*, *tin4*, and *tin5* [44]. The Tin1 proteins form a group of weakly related *U. maydis* effectors with amino acid identities of 34–48%. Deletion mutants lacking all five genes show slightly reduced tumor formation on maize leaves and lead to upregulation of endo-chitinases, SA-binding proteins, the apoplastic peroxidase POX12, and an NBS-LRR class disease resistance gene in maize, indicating that the effectors contribute to the suppression of basal host immunity [44]. However, individual contribution to virulence of the five Tin1-family proteins could not be ascertained, putatively because their individual effects on virulence were too small to be reliably detected [44].

Deletion of diversity region 19-1 in *S. reilianum* indicated that the cluster contained virulence effectors, both in the large first part (19A1) as well as in the small second part (19A2) [43]. These included: *sr10079*, one of the three *S. reilianum* homologs of *tin4*, *sr10059*, and *sr10057*; the *S. reilianum* homolog of *tin2* but not *sr10060*; the *S. reilianum* homolog of *tin3*, *sr10073*; the *S. reilianum* homolog of *tin5*, or *sr10075* and *sr10077,* and; the other two *S. reilianum* homologs of *U. maydis tin4* [43]. This suggests the evolution of species-specific functions for the individual effectors encoded in the divergence region 19-1 of *U. maydis* and *S. reilianum*. In this study, we dissected the large first part (19A1) of the divergence region 19-1 of *S. reilianum* to identify one gene of a three-membered gene family with similarity to *tin1* of *U. maydis*, *sr10050*, as a virulence effector gene that we named *vag2* (*virulence-associated gene 2*). We found that the secreted virulence effector Vag2 of *S. reilianum* is transcriptionally upregulated during biotrophic growth and suppresses plant defense through interfering with SA-dependent gene expression by directly interacting with the chorismate mutase 2 (ZmCM2) of its host plant maize.

## 2. Materials and Methods

### 2.1. Strains and Growth Conditions of S. reilianum and Maize, Symptom Scoring

The maize line (*Zea mays*) cultivar ‘Gaspe Flint’ was used throughout this study and was originally supplied by Regine Kahmann, Marburg, Germany. Kernels were sown in Type T Fruhstorfer soil and grown in a glasshouse with a 15-h day period at 28 °C and 50% relative humidity, and a 9-h night period at 22 °C and 60% relative humidity. Seven-day-old seedlings were used for plant inoculation, and symptoms were evaluated after the emergence of inflorescences at 7 weeks post-inoculation as described [43].

For plant inoculation experiments, compatible *S. reilianum* f. sp. *zeae* wild-type strains SRZ1_5-2 and SRZ2_5-1 [6] and their deletion derivatives were used. The strains used in this study are listed in Appendix A. All *S. reilianum* strains were stored at −80 °C in 25% glycerol, and freshly streaked on potato dextrose (PD) agar medium (BD, Heidelberg, Germany) and cultivated for 3 days at 28 °C prior to any experiments. Single colonies were used to inoculate 2-mL cultures of YEPSlight medium (1% tryptone, 1% yeast extract, 1% sucrose) and grown for 8 h at 28 °C with shaking at 200 rpm. For plant inoculation, the liquid cultures were used to inoculate 50 mL of PD medium (BD, Heidelberg, Germany) that were incubated overnight until the optical density at 600 nm (OD_600_) reached a value of 0.6. The cultures were pelleted by centrifugation, and cell pellets were resuspended in water to a calculated OD_600_ of 2.0. Cultures of two mating-compatible strains were mixed at a 1:1 (*v*:*v*) ratio.

### 2.2. Generation of S. reilianum Gene Deletion and Complementation Strains

Gene deletion strains were generated by the replacement of the gene or gene region of interest with the hygromycin resistance cassette by double homologous recombination. The hygromycin resistance cassette was excised from pBS-hhn as a 1.8-kb *Sfi*I fragment (pBS-hhn; [45]). The right and left flanking regions of the regions of interest were amplified by PCR, digested with the restriction enzyme *Sfi*I and ligated to the resistance cassette. Nested primers were used to amplify the complete deletion construct that was used for the transformation of wild-type strains. For the generation of complementation strains, the constructs were similarly generated but contained the nourseotricin resistance cassette of pBS-Nat [45] and the complete genes to be introduced (*sr10050*, *sr10051*, or *vag2-GFP*) including the native promoter and terminator sequences. Amplified complementation constructs were used to transform the ΔA8 or *Δvag2* deletion strains replacing the hygromycin resistance cassette by chemical transformation of protoplasts as described [43]. Up to three independently generated and PCR- and Southern blot-validated strains for each mating background were used for plant inoculation experiments. The strains used in this work are listed in Appendix A. Primers used for construct generation are listed in Appendix A.

### 2.3. Genomic DNA and RNA Isolation, and qRT-PCR Analysis

Genomic DNA of *S. reilianum* was isolated according to a modified protocol described by [46]. In brief, overnight cultures were pelleted by centrifugation in the presence of glass beads (200 μL) at 13,000 rpm for 5 min. Supernatants were aspirated and pellets were frozen at −20 °C for at least 20 min. The cells were resuspended in *Ustilago* lysis buffer (500 µL; Tris-HCl pH 7.5 at 50 mM, sodium ethylene tetra acetate (EDTA) at 50 mM, sodium dodecyl sulfate (SDS) at 1%) and a 1:1 mixture of phenol:chloroform (600 µL). After 15 min Vibrax shaking at maximum speed, the cell debris was pelleted by centrifugation at 13,000 rpm for 25 min at room temperature, and the upper phase was transferred to a fresh tube. The DNA was precipitated with the addition of ethanol (96%; 1 mL) and pelleted by centrifugation at 13,000 rpm for 15 min. The supernatant was aspirated and the pellet was left to dry at room temperature. The DNA pellet was resuspended and solved in TE buffer (50 µL; Tris-HCl pH 8.0 at 10 mM, EDTA at 1 mM) containing 20 mg/mL RNase A at 55 °C for 15 min and stored at −20 °C.

RNA was extracted with Trizol reagent (Invitrogen, Waltham, MA, USA) and purified with the RNeasy Plant Mini Kit (Qiagen, Hilden, Germany). The cDNA was synthesized using the RevertAid H Minus First Strand cDNA Synthesis Kit (Fermentas, St. Leon-Rot, Germany). The cDNA was used for qPCR. Reactions for qPCR contained cDNA (20–50 ng), 1× Taq Buffer (Bioline, Luckenwalde, Germany), MgCl_2_ (3 mM), dNTPs (0.1 mM), forward and reverse primers (200 nM each), SYBR Green solution (0.1 vol of a 1:10,000 dilution), and Bio-Taq DNA Polymerase (Bioline, Luckenwalde, Germany, 0.25 U) in a total volume of 25 µL. PCR was performed using the CFX Connect™ Real-Time PCR Detection System (Bio-Rad, Feldkirchen, Germany). Cycling parameters were the same for all primers: 95 °C for 6 min, followed by 40 cycles of 95 °C for 30 s, 60 °C for 30 s, 72 °C for 1 min, plate read step, then product melting curve 55–95 °C. The results were analyzed with CFX Manager 3.0 (Bio-Rad). Transcript levels of *vag2* were determined relative to those of the glyceraldehyde 3-phosphate dehydrogenase (GAPDH) gene from *S. reilianum* (*sr10940.2*). For analysis of the maize *PR1* (AC205274.3_FG001) and *PR5* (GRMZM2G402631) genes, transcript levels were calculated relative to that of actin (*ACT1*) (GRMZM2G126010).

### 2.4. Bimolecular Fluorescence Complementation and Fluorescence Microscopy

The open reading frames of Vag2 lacking its signal peptide sequence (Vag2∆SP) and of ZmCM2 were cloned in the pDONR201 entry vector using Gateway cloning and introduced in their respective destination vectors (pE-SPYCE-GW and pE-SPYNE-GW) [47]. The N terminus of Vag2∆SP was fused to the C-terminal half of YFP (SPYCE), whereas the N-terminal half of YFP was fused to the N terminus of ZmCM2 (SPYNE). The plasmids were introduced in the *Agrobacterium tumefaciens* strain GV3101:pMP90RK [48]. For transient expression, *A. tumefaciens* strains were grown in the presence of appropriate antibiotics to an OD_600_ of 1.0 and used for infiltration of 4- to 5-week-old *Nicotiana benthamiana* leaves. Three days after infiltration, the leaves were analyzed for YFP fluorescence using a Leica DM 6000B fluorescence microscope. For detection of Vag2-GFP, 1-cm leaf and ear samples were covered with PBS (NaCl, 8 g/L; KCl, 0.2 g/L; Na2HPO4, 1.44 g/L; KH2PO4, 0.24 g/L) and used for direct fluorescence analysis.

### 2.5. Yeast Two-Hybrid Screening

For yeast two-hybrid screening and targeted yeast two-hybrid experiments, the Matchmaker^TM^ Gold Yeast Two-Hybrid system was used. For library screening, the bait protein was expressed as a GAL4 DNA-binding domain (BD) fusion to the Vag2 open reading frame lacking the signal peptide sequence (pGBKT7-Vag2ΔSP), and introduced into the Y2HGold strains to give rise to the Y2HGold-BD-Vag2ΔSP strain. The expression of BD-Vag2ΔSP was verified by Western blot using a monoclonal antibody against the c-myc tag. Prey proteins were expressed as GAL4 activation domain (AD) fusion proteins. An existing cDNA library of *S. reilianum*-inoculated maize [49] was used, and library plasmids were introduced into the Y187 strain. The Y2HGold-BD-Vag2ΔSP strain and the Y187 strain containing the prey library plasmids were mated for library screening as described [49]. For the targeted yeast two-hybrid experiment, the Y2HGold-BD-Vag2ΔSP strain was mated with Y187 containing plasmid IP1 that was recovered from the cDNA library and contained maize chorismate mutase-derived sequences in fusion to the GAL4 activation domain (AD-ZmCM2). The yeast two-hybrid experiments were performed as described in the Matchmaker™ Gold Yeast Two-Hybrid System User Manual from Clontech. Strains expressing tumor suppressor p53 (AD-P53) and the strong interaction partner SV40 T-antigen (BD-T) were used as a positive control, whereas strains expressing Vag2 (BD-Vag2ΔSP) or Lamin (AD-LAM) and T-antigen (BD-T) were used as negative controls.

### 2.6. Yeast Secretion Trap Assay

The open reading frame of Vag2 was amplified using oSD23 and oSD24 (Appendix A) and was cloned in the pYST-1 vector in frame with the invertase gene. vag2-pYST-1 and pYST-1 were used to transform the *S. cerevisiae* SEY6210 strain that is auxotrophic for leucine and cannot utilize sucrose as a carbon source. The experiment was conducted as described [50].

### 2.7. Metabolite Analysis

Maize leaves inoculated with H_2_O (Mock), *S. reilianum* wild-type, or *∆**vag2* deletion strains were collected at 6 days post-inoculation (dpi), and young ears of inoculated plants at 31 dpi. Leaf sections (around 2 cm in size) from above (Up) and below (Down) the inoculation site were collected separately for each inoculation and deep-frozen in liquid nitrogen. These samples were used for the analysis of SA by LC-MS as described before [51], and of metabolites by gas chromatography coupled-mass spectrometry (GC-EI/TOF-MS). For SA measurement, three biological replicates per treatment with a second group of technical replicates were analyzed, with each biological replicate containing a pool of 8–10 leaves or ears from different plants. For metabolite analysis, 7–8 leaves of independent plants were analyzed as biological replicates.

In detail, multi-targeted metabolite profiling by gas chromatography coupled to electron impact ionization time-of-flight mass spectrometry (GC-EI/TOF-MS) was used for the analysis of a metabolite fraction enriched for small and primary metabolites [52]. Soluble metabolites were extracted by chloroform:methanol:water with added internal standard, ^13^C_6_–sorbitol, from deep-frozen tissue powder that was produced by an oscillating ball mill [52]. Samples were kept frozen at −80 °C until liquid extraction. Extracts were dried by vacuum concentration and stored at −20 °C until GC-MS analysis. Chemical derivatization and retention index calibration for EI/TOF-MS analysis were as described previously [52]. GC-EI/TOF-MS analysis was performed using an Agilent 6890N24 gas chromatograph (Agilent Technologies, Waldbronn, Germany) connected to a Pegasus III time-of-flight mass spectrometer (LECO Instrument GmbH, Möchengladbach, Germany), with splitless injection onto a Varian FactorFour capillary column (VF-5 ms) of 30 m length, 0.25 mm inner diameter, and 0.25 µm film thickness (Varian-Agilent Technologies, Waldbronn, Germany). Chromatograms were acquired, visually controlled, baseline corrected, and exported in NetCDF file format using ChromaTOF software (Version 4.22; LECO Instrument GmbH, Möchengladbach, Germany). Compounds were identified by mass spectra and retention time index matching to the reference collection of the Golm Metabolome Database [53] with manual supervision using TagFinder software [54]. Guidelines for manually supervised metabolite identification were the presence of at least three specific mass fragments per compound and a retention index deviation of less than 1.0% [55]. Metabolite intensities, i.e., peak heights of arbitrary units, were normalized by sample fresh weight and abundance of internal standard (^13^C_6_–sorbitol). Significance testing was by Student’s *t*-test (*p* < 0.05).

## 3. Results

### 3.1. Diversity Region 19-1 of S. reilianum Contains the Virulence Gene vag2

The first part of the divergence region 19-1 (A1) comprises 26 genes (Figure 1). The deletion of all 26 genes results in reduced virulence of *S. reilianum* on maize [43]. At least two effectors encoded in this region (sr10057, sr10059) contribute incrementally to total virulence [43]. To identify putative additional virulence-affecting effectors of this region, we generated and tested sub-deletion mutants of cluster 19-1. To this end, we first created deletion strains lacking region A3 (ΔA3; Figure 1) which comprises the first 13 genes of cluster 19-1. Next, we generated five independent deletion strains in each of the two mating-compatible wildtype strains SRZ1_5-2 and SRZ2_5-1, by replacing the 13-gene region with a hygromycin resistance cassette and confirmed the deletion strains by Southern blot. We randomly selected two strain pairs (ΔA3#1, ΔA3#2; Appendix A) and tested virulence. Virulence of the ΔA3 deletion strains was clearly reduced relative to wild type, but nearly identical to virulence of ΔA1 deletion strains (Figure 2A, Appendix A). Therefore, we concluded that the A3 region contains important effectors contributing to the virulence of *S. reilianum*.

Next, we generated strains lacking regions A4 and A5 that comprise the first and the last six genes each of region A3 (Figure 1). We generated four independent deletion strains in each of the two mating-compatible wildtype strains SRZ1_5-2 and SRZ2_5-1 by replacing the A4 region with a hygromycin resistance cassette and confirmed the deletion strains by Southern blot. We also generated three and two deletion strains, respectively, in the two mating-compatible wildtype strains SRZ1_5-2 and SRZ2_5-1 by replacing the A5 region with a hygromycin resistance cassette and confirmed the deletion strains by Southern blot. We randomly selected one strain pair each and tested the virulence of the deletion strains in three independent experiments. Virulence of the ΔA4 deletion strains was significantly reduced relative to wildtype and ΔA3 deletion strains, while virulence of the ΔA5 deletion strains was nearly identical to that of the ΔA3 deletion strains (Figure 2B, Appendix A).

Because we had previously identified two individual effector genes of region A5 (*sr10057*, *sr10059*) as contributing weakly to the overall virulence of *S. reilianum* [43], we did not dissect the A5 region any further. Instead, we decided to generate two additional subcluster deletions lacking respectively the first (A8) and the last (A9) three genes of the region A4 (Figure 1). We independently generated at least three individual deletion strains in each of the two mating-compatible wildtype strains SRZ1_5-2 and SRZ2_5-1 by replacing either the A8 or the A9 region with a hygromycin resistance cassette and confirmed the deletions by Southern blot. We then tested the virulence of three pairs of independently generated deletion strains lacking either the three genes of region A8 or of A9. Virulence of strains lacking either the A8 or the A9 region was clearly reduced but the A8 region seemed to have a stronger effect on virulence (Figure 2C, Appendix A).

Of the three genes encoded in region A9, only one of them (*sr10053*, Figure 1) was detectable in a preliminary RNAseq experiment of *S. reilianum*-infected leaf tissue. To test whether this gene contributed to the virulence of *S. reilianum* on maize, we generated three independent deletion strains (Δ53, Appendix A) in each of the two mating-compatible wildtype strains SRZ1_5-2 and SRZ2_5-1 by replacing the *sr10053* gene with a hygromycin resistance cassette and confirmed the deletion strains by Southern blot (not shown). We then tested the virulence of three pairs of independently generated deletion strains. Virulence of the Δ53 gene deletion strains was not significantly different from wild type (Figure 2C, Appendix A). It could be that both or either one of the genes *sr10054* or *sr10055* contribute to virulence, or that each of the three genes of region A9 contributes only marginally so that all three genes need to be deleted to observe an effect on virulence.

Lack of region A8 of cluster 19-1 decreased virulence of *S. reilianum* (Figure 2C). To find out which of the three genes contributes to virulence, we independently generated at least four individual deletion strains for each of the two mating-compatible wildtype strains SRZ1_5-2 and SRZ2_5-1 by replacing the open reading frames of either *sr10050*, *sr10051,* or *sr10052.2* (Figure 1) with a hygromycin resistance cassette and confirmed the deletions by Southern blot. We randomly selected three strain pairs each (Δ50, Δ51, Δ52; Appendix A) and tested virulence. Deletion of *sr10050* and *sr10051* but not of *sr10052.2* resulted in a significant decrease in overall virulence, with *sr10051* affecting only disease incidence but not disease severity (Figure 2D, Appendix A). This indicates that the lack of *sr10052.2* may not contribute to the reduced virulence of the ΔA8 deletion strains.

To find out whether the lack of *sr10050* or *sr10051* or both contribute to the reduced virulence of the ΔA8 deletion strains, we introduced the open reading frame of either *sr10050* or *sr10051* together with their native promoters and terminators and a nourseothricin resistance cassette into two mating-compatible ΔA8 deletion strains by replacing the hygromycin cassette. Successful gene integration and replacement of the hygromycin resistance cassette were confirmed by PCR and Southern blot (not shown). Southern blot also confirmed single integration of the replacement construct. At least four independent complementation strains were obtained for each of the two mating-compatible ΔA8 deletion strains and for each of the complementation constructs. The virulence capacity of the complementation strains was tested by leaf-whorl inoculation of maize seedlings with a mixture of two mating-compatible complementation strains (Appendix A). The re-introduction of *sr10050* completely restored the virulence of ΔA8 deletion strains (Figure 2D, Appendix A). However, the re-introduction of *sr10051* did not significantly affect the virulence of the ΔA8 deletion strains (Figure 2D, Appendix A). These results indicate that of the genes encoded in the region A1 of the divergence region 19-1 (Figure 1) of *S. reilianum*, *sr10050* contributes most to the virulence of *S. reilianum* on seedling leaves. Therefore, we named *sr10050* as *vag2* (*virulence-associated gene 2*).

### 3.2. S. reilianum vag2 Is Transcriptionally Upregulated during Fungal Biotrophic Growth

To determine at which stage—before or during plant colonization—*vag2* is expressed, we measured relative *vag2* mRNA abundance by quantitative RT-PCR. To this end, we isolated total RNA from the prepared mating mixture of *S. reilianum* wildtype strains SRZ2_5-1 and SRZ1_5-2 immediately before inoculation, and from *S. reilianum*-colonized maize tissues (leaves and ears) at different time points (2, 4, 6, and 9 dpi for leaves, and 31 dpi for ears) after inoculation with the mating mixture. RT-PCR was done with a *vag2*-specific primer pair (Appendix A) and a primer pair against the *S. reilianum* glyceraldehyde 3-phosphate dehydrogenase (GAPDH) gene (*sr10940.2*) (Appendix A) as control. The expression of *vag2* in inoculated leaves and ears was calculated relative to the expression of the *GAPDH* gene and relative to the expression of *vag2* in axenic culture. The *vag2* gene was strongly expressed throughout the complete biotrophic phase. The abundance of the *vag2* mRNA was increased by 30–260 times in leaves at 2, 4, 6, and 9 dpi relative to the axenic culture and increased to more than a thousand times in young ears at 31 dpi (Figure 3A). The notable increase in expression of *vag2* during the colonization of maize suggests a crucial role of *vag2* throughout the biotrophic phase of *S. reilianum*.

### 3.3. vag2 Deletion Mutants Show Reduced Systemic Spread in Maize

To evaluate the effect of *vag2* deletion on fungal proliferation in maize, we quantified fungal DNA in leaves at 3 dpi and in nodes at 14 dpi after inoculation of maize seedlings with either wildtype or *Δ**vag2* deletion strains. Total genomic DNA was isolated from a pool of 10 plant samples of 10 different plants per replicate, collecting 3-cm pieces of inoculated leaves below the inoculation site or complete nodes at the base of the inoculated leaf, and used for quantitative PCR using primer pairs for detection of the fungal GAPDH and the maize actin genes (Appendix A). We measured the amount of fungal relative to plant DNA and compared the values for wildtype and *Δ**vag2* deletion strains. Relative to the wildtype strains, *Δvag2* deletion strains reproducibly showed a higher proliferation density in leaf samples while showing a significantly lower density in nodes of infected maize (Figure 3B). This indicates that Vag2 may have a role in directing fungal growth towards systemic tissues since it has an inhibitory effect on fungal proliferation in leaves and supports fungal spread into nodes of infected maize seedlings.

### 3.4. Vag2 Has a Functional Secretion Peptide

To investigate whether the predicted secretion peptide at the N terminus of Vag2 is functional, a yeast secretion trap assay was performed. The Vag2 coding sequence was cloned in frame with the C-terminal part of the invertase gene lacking the native secretion signal peptide sequence in the vector pYST-1. Both the empty vector (pYST-1) as well as the vector containing *vag2* (pYST-vag2) were used to transform the *S. cerevisiae* strain SEY6210 that lacks the native *SUC2* invertase gene. Transformants were tested for growth on SD minimal media lacking leucine for plasmid selection and containing either glucose or sucrose as a carbon source. Yeast strains containing pYST-vag2 grew well on both carbon sources, whereas transformants containing the empty vector only grew on glucose-containing media (Figure 4A). This suggests that the predicted secretion peptide of Vag2 is functional and should lead to the secretion of the protein during the biotrophic growth of the fungus.

To corroborate this suggestion, we introduced a *vag2-GFP* fusion into two mating-compatible Δ*vag2* deletion strains at the *vag2* native locus generating Δ*vag2+vag2:GFP* strains. Correct construct integration was verified by PCR and Southern blot (not shown). Three independent transformants each were used for three independent maize inoculation experiments. Maize seedlings were inoculated with wildtype, Δ*vag2,* and Δ*vag2+vag2:GFP* strains, and inoculated leaves (1, 2, and 3 dpi) as well as developing ears (31 dpi) were collected for fluorescence microscopic analysis, and disease symptoms were analyzed at 8 weeks post-inoculation. Virulence evaluation suggested that the Vag2-GFP fusion protein does not functionally replace Vag2 (Figure 4B), possibly because the GFP tag interfered with protein interaction, folding, or localization of Vag2. In spite of lacking in planta activity of the Vag2-GFP fusion protein, fluorescence microscopic analysis clearly detected GFP fluorescence in and around the fungal hyphae in both maize leaves and ears (Figure 4C). Together, these results indicate that Vag2 contains a functional secretion signal peptide and is secreted by hyphae of *S. reilianum* during the colonization of maize.

### 3.5. Interaction Partners of Vag2 Are Involved in Various Plant Processes

To identify putative Vag2-interacting proteins, we screened an available normalized yeast two-hybrid (Y2H) library generated from pooled RNA isolated from *S. reilianum*-colonized leaves and ears as well as from a mating mixture of *S. reilianum* wildtype strains [49]. To this end, we cloned *vag2* lacking sequences encoding its secretion signal peptide in the pGBKT7 vector and used the resulting pGBKT7-vag2ΔSP to transform the Y2HGold strain. The Y2HGold strain containing pGBKT7-vag2ΔSP was mated with the Y187 yeast strain containing the cDNA library. About 10^6^ independent diploid colonies were obtained and selected on an SD medium lacking leucine and tryptophan to ensure plasmid maintenance. Of the 10^6^ colonies, only 158 grew on SD medium also lacking adenine and histidine and containing 3-amino-1,2,4-triazole (3-AT). The containing 158 plasmids were recovered and their inserts sequenced. Sequence analysis resulted in 62 unique sequences. The sequences were compared to the sequence databases at NCBI (BLASTN [56]) and at MaizeGDB [57]. The sequences could be mapped to 43 genes encoding different plant proteins. No putative interaction partner was of fungal origin. The putative plant Vag2-interaction proteins were involved in metabolism, DNA binding, protein folding, signaling, or nuclear processes (Table 1). Of the 43 identified Vag2-targets, five were retrieved more than five times: The *Z. mays* chorismate mutase 2 (24x), a ThiC superfamily protein (11x), a pleckstrin homology domain-containing protein (10x), an unknown protein containing a NOT2,3,5 domain (7x), and a citrate synthase family protein (6x). Of all putative Vag2-interacting proteins, the *Z. mays* chorismate mutase 2 (ZmCM2) was retrieved most often. Therefore, we decided to investigate whether ZmCM2 interacts with Vag2.

### 3.6. Vag2 Interacts with the Maize Chorismate Mutase 2 (ZmCM2)

We first checked the putative interaction between ZmCM2 and Vag2 by retransforming the Y187 strain with one of the retrieved plasmids (pIP1) containing the ZmCM2-derived sequence. The resulting strain was mated with the Y2HGold strain expressing the GAL4-binding domain fused to Vag2 lacking its signal peptide and selected for diploids. The resulting diploids grew well on penta-selective media (SD minimal medium lacking tryptophan, leucine, adenine, and histidine and containing aureobasidine A) confirming that the cloned sequences interacted in the Matchmaker Gold Yeast Two-Hybrid System (Figure 5A).

To confirm the suggested interaction between ZmCM2 and Vag2 and also find out whether they interact when expressed in plant tissues, we conducted a bimolecular fluorescence complementation assay (BiFC [47]). To this end, we cloned *vag2* lacking sequences encoding its signal peptide in fusion with the C-terminal half of the yellow fluorescent protein (YFP; C-YFP-Vag2ΔSP). We amplified the complete open reading frame of ZmCM2 from maize cDNA and cloned it as a fusion to the N-terminal half of YFP (N-YFP-ZmCM2). Both fusion proteins were co-expressed in *Nicotiana benthamiana* following *A. tumefaciens*-mediated transient transformation. Three days after bacterial infiltration, a strong YFP fluorescence signal was detected in epidermal cells of tobacco leaves that expressed C-YFP-Vag2ΔSP and N-YFP-ZmCM2 (Figure 5B) which was not visible when either fusion protein was co-expressed with the N- or C-terminal parts of YFP, respectively (Figure 5B). This result confirmed that Vag2 and ZmCM2 interact when expressed in planta. In addition, the fluorescence signal was visible both in the cytoplasm and in the nucleus of transformed cells (Figure 5B), suggesting that Vag2 and ZmCM2 form stable complexes in planta that localize to both plant compartments.

### 3.7. S. reilianum Δvag2 Deletion Strains Slightly Increase the SA Level in Colonized Tissue and Induce SA-Related Defense Gene Expression in Maize

Chorismate mutase is a pivotal shikimate biosynthetic enzyme that catalyzes chorismate conversion into prephenate. Additionally, chorismate is also a precursor for the biosynthesis of SA [58]. Since ZmCM2 interacts with Vag2, we speculated that this interaction could interfere with the biosynthesis of SA in the host. Therefore, we collected leaf and ear samples for quantification of the SA content in maize tissues. To compare colonized versus non-colonized tissues, we collected two-centimeter leaf samples below the inoculation site at 6 dpi, where fungal proliferation is abundant. In addition, we collected young ears of about 1.5 cm in size at 31 dpi of *Z. mays* cv. ‘Gaspe Flint’ inoculated with either de-ionized water, *S. reilianum* wildtype, or *Δ**vag2* deletion strains. The collected samples were used for the analysis of SA concentrations using high-performance liquid chromatography coupled to mass spectrometry (HPLC-MS). In both leaf and ear samples, the SA concentration was reproducibly but not significantly higher in samples inoculated with *Δvag2* deletion strains than with wildtype strains (Figure 6A,B).

To confirm that deletion of *vag2* influences the SA levels in *S. reilianum*-colonized tissues, we investigated the transcript amount of the two SA-indicator genes, PATHOGENESIS-RELATED GENE 1 (*PR1*; AC205274.3_FG001) and PATHOGENESIS-RELATED GENE 5 (*PR5*; GRMZM2G402631) by quantitative RT-PCR. Maize leaves and ears inoculated with water, *S. reilianum* wildtype, and *Δ**vag2* deletion strains were collected at 6 and 31 dpi, respectively. Total RNA was isolated from collected samples and used to synthesize cDNA, which was used as a template for quantitative PCR. Compared to the samples colonized with *S. reilianum* wildtype strains, *PR1* expression in leaves and ears colonized by *Δvag2* deletion strains was five and nine times upregulated, respectively (Figure 6C,D). The transcript levels of *PR5* in leaves and ears colonized with *∆vag2* deletion strains were eight and 1.8 times higher, respectively, when compared to tissues colonized by wildtype strains (Figure 6E,F). Upregulation of the SA-induced defense genes *PR1* and *PR5* in maize tissues inoculated with *Δvag2* deletion strains suggests that Vag2 might suppress or interfere with the SA-related defenses in the host during *S. reilianum* colonization.

### 3.8. Metabolite Flux Is Redirected from SA Generation to Aromatic Amino Acid Accumulation

To check whether upregulation of SA biosynthesis in leaves infected with *S. reilianum* lacking *vag2* resulted in downregulation of other metabolites, we collected leaf samples below the inoculation site (tissue that is generally colonized) and samples above the inoculation point (tissue that is generally not or only weakly colonized) and measured metabolite concentration by gas chromatography-coupled mass spectrometry (GC-MS). We focused on the analysis of metabolites known or suspected to be involved in SA or SA-precursor metabolism. SA is thought to be generated via two pathways in plants, (1) via the shikimate pathway, chorismate and isochorismate, or (2) via phenylalanine, cinnamate, and benzoate [59]. Route 1 has been identified as the predominant SA-generation route in response to pathogen infection in Brassicaceae plants [58]. The two routes share chorismate as a precursor and are thus linked via the action of the chorismate mutase that is thought to lower SA precursor concentration by converting chorismate to prephenate [60]. Prephenate is a precursor for the generation of the aromatic amino acids tyrosine and phenylalanine [61] that is solely the source of SA biosynthesis via route 2.

We observed detectable concentrations of shikimate, benzoate, phenylalanine, tyrosine, and the tyrosine-derived metabolite tyramine but not of SA whose concentrations were below the detection limit in these samples. The biggest concentration differences could be observed in the systemic non-colonized tissue above the inoculation site for tyrosine, phenylalanine, and benzoate. The concentrations of shikimate, tyrosine, tyramine, and phenylalanine but not of benzoate were increased in *S. reilianum*-colonized tissues relative to uninfected control tissues (Figure 7A). In samples inoculated with *Δvag2* deletion strains, the concentrations of tyrosine, tyramine, and phenylalanine were decreased in at least one of the samples collected from above or below the inoculation site relative to wildtype-inoculated tissues (Figure 7A). In contrast, the concentrations of benzoate were lowest in samples inoculated with wildtype strains (Figure 7A). Because the concentration of cinnamate was below the detection limit, we investigated the concentration of 4-hydroxy-cinnamate, caffeate, and ferulate. Cinnamate can be converted to 4-hydroxy-cinnamate by cinnamate-4-monooxygenase which is encoded by a small gene family in maize [62]. Alternatively, 4-hydroxy-cinnamate could also be generated by the action of the tyrosine ammonia-lyase or phenlyalanine ammonia-lyase directly from tyrosine or phenylalanine [63]. On the other hand, 4-hydroxy-cinnamate can also be converted to caffeate and ferulate, all-important precursors for lignin biosynthesis [64]. None of these metabolites (4-hydroxy-cinnamate, caffeate, or ferulate) were increased in wildtype-infected samples (Figure 7A). Together, these results indicate that also in *S. reilianum*-colonized maize leaf tissue, SA biosynthesis does not follow the phenylalanine, cinnamate, and benzoate pathway. In addition, increased SA levels in leaves of plants inoculated with *Δvag2* deletion strains relative to wildtype strains are paired with reduced levels of tyrosine, tyramine, and phenylalanine, indicating that Vag2 positively influences metabolite flux from SA to tyrosine and phenylalanine (Figure 7B).

## 4. Discussion

Genome comparison between *S. reilianum* and *U. maydis* led to the identification of divergence regions containing gene clusters of low sequence conservation [2]. The largest divergence region (19-1) has been identified as necessary for full virulence in both *U. maydis* and *S. reilianum* [25,43]. In *U. maydis*, the virulence-contributing genes have been dissected, and eight genes contributing to virulence have been identified (Figure 1; [44]).

The tumor-inducing genes *tin4* and *tin5* of *U. maydis* have homologs in the A2 region of 19-1 of *S. reilianum* that has been shown to significantly contribute to virulence [43]. The homologs of the *tin2* and *tin3* genes of *U. maydis* are located in the A5 region in the cluster 19-1 of *S. reilianum*. This region contains two genes that were previously identified as contributing to the overall virulence of *S. reilianum*: *sr10057*, a homolog of *tin2*, and *sr10059* which is unrelated to *tin3* [43]. Their contribution to virulence seems to be small since the ΔA5 deletion strains are nearly as virulent as the ΔA3 deletion strains. Alternatively, their function depends on the presence of at least one of the proteins encoded in the A3 region. It is not known how the identified virulence-contributing proteins of *S. reilianum* function and if they form protein-protein complexes. Recently, an effector-protein complex involving the seven members Pep1, Stp1, Stp2, Stp3, Stp4, Stp5, and Stp6 has been found in *U. maydis*. The complex is formed during biotrophic growth and may be involved in effector translocation into plant cells [65]. A systematic yeast two-hybrid interaction screen of *U. maydis* effectors revealed extensive complex formation capacity, both by homo- as well as heterodimerization [66].

A five-gene cluster of related *U. maydis* genes named *tin1-1*, *tin1-2*, *tin1-3*, *tin1-4,* and *tin1-5* was shown to be important for the virulence of *U. maydis* [44]. A *tin1-1* to *tin1-3* subcluster deletion led to decreased virulence, just as a *tin1-3* to *tin1-5* deletion, but single deletion of *tin1-3* did not lead to a measurable reduction in virulence. This led the authors to propose that the individual contribution of each of the five *tin1* genes to virulence could be too small for detection [44]. In *S. reilianum*, there are three homologs of the *U. maydis tin1* genes, all located in the A8 region of divergence region 19-1: *vag2*, *sr10051,* and *sr10052* (Figure 1). Deletion of the A8 region led to reduced virulence. When individually deleting each of the three genes of region A8 in *S. reilianum*, deletion of *vag2* led to decreased disease incidence and decreased disease severity, while deletion of *sr10051* decreased disease incidence but not disease severity, and deletion of *sr10052.2* did not affect disease incidence and positively affected disease severity. This indicates that each of the three genes individually affects virulence and that their role in virulence is independent of the presence of the other two genes. Of the five *U. maydis tin1* genes, two were positive in the yeast two-hybrid effector interaction screen, and for one of them, *tin1-2*, three putative interaction partners were retrieved [66]. Since these are identified as unrelated proteins, there is also no indication of complex formation between different Tin1 proteins in *U. maydis*.

The first identified avirulence effector (UhAvr1) of *Ustilago hordei*, a related barley smut pathogen, was found to reside in the *U. hordei* version of cluster 19 [67]. UhAvr1 shows the highest similarity to Tin1-2 of *U. maydis*. The next closest similarity is to Tin1-3 and Tin1-6 of *U. maydis* which, respectively, are most closely related to Vag2 and Sr10052.2 of *S. reilianum* [67]. In contrast to *vag2* which was expressed throughout the biotrophic phase with the highest values in young ears (Figure 3A), Uhavr1 was expressed only in the early stages of the biotrophic phase [68] suggesting specific roles of the two proteins in their respective pathosystems. UhAvr1 was shown to lead to complete immunity in barley cultivars carrying the resistance gene Ruh1, which triggers hypersensitive cell death (HR). In contrast, in susceptible plants, the protein contributes to fungal virulence by suppressing general defense responses including nonhost/PTI and ETI invoked by various pathogens [68].

Deletion of *vag2* had the strongest effect on virulence of *S. reilianum* of all tested individual gene deletions of the A1 region. The gene is expressed throughout the biotrophic phase (Figure 3A) but is not essential for the growth and proliferation of *S. reilianum* since deletion strains can penetrate and spread through the host tissues. DNA quantification indicated that *Δvag2* deletion strains multiplied more in the inoculated leaf than in wildtype strains but showed a somewhat lower presence in node tissues (Figure 3B). The transition from local proliferation in the inoculated leaf to systemic spread via the vascular bundles into the nodes is an important step for successful colonization of the host plant. On sorghum, these *S. reilianum* strains can multiply well in the leaf but do not arrive in the nodes, and inoculated plants stay healthy [7]. Possibly, Vag2 is important for lowering the plant defense responses in tissues needed for systemic spread.

In support of this hypothesis, we reproducibly found an increased level of SA and a significantly increased expression of the SA-indicator genes *PR1* and *PR5* in leaves and ears of plants inoculated with *Δvag2* deletion strains than with wildtype strains (Figure 6). Pathogen-induced SA generation in plants is thought to be via the shikimate-chorismate-isochorismate pathway and not via the alternative phenylalanine-cinnamate-benzoic acid pathway. For Brassicaceae, the shikimate-chorismate-isochorismate pathway was shown to depend on the action of the amidotransferase PBS3 [69,70]. Whether microbe-induced SA generation in maize follows the same or an alternative pathway still needs to be determined. Metabolite analysis indicated that maize infection with *S. reilianum* does not lead to increased levels of benzoate, 4-hydroxy-cinnamate, caffeate, or ferulate, which supports the notion that *S. reilianum*-induced SA generation in maize is not via the phenylalanine-cinnamate-benzoate pathway. The increased level of SA in tissues colonized with *Δvag2* deletion strains corresponded to a lower level of the metabolites tyrosine, tyramine, and phenylalanine in tissues inoculated with *Δvag2* deletion strains than with wildtype strains (Figure 7A). This indicates that Vag2 is necessary for redirecting metabolite flow away from SA towards the metabolites tyrosine, tyramine, and phenylalanine.

The SA precursor chorismate can be converted into prephenate by the chorismate mutase. Maize contains two chorismate mutases, ZmCM1 which is predicted to locate in the chloroplasts, and the putative cytosolic ZmCM2 [22]. We found that ZmCM2 interacts with Vag2, both as the best hit in a yeast two-hybrid screen, and when expressed as fusions to the N- and C-terminal halves of YFP in planta (Figure 5). It is unclear which effect the interaction of Vag2 with ZmCM2 has on the function of the maize chorismate mutase. However, our data speak against an inhibiting function since we observed an increased level of the prephenate-metabolic products tyrosine and phenylalanine in the presence of Vag2. In our opinion, our data is best explained by a subtler effect on enzyme function, for example, by making the ZmCM2-catalyzed interconversion of chorismate and prephenate irreversible in the direction of prephenate generation (Figure 7B). It has been shown for *U. maydis* that increasing the activity of chorismate mutase by secretion of a fungal chorismate mutase into the colonized tissue leads to a lowering of the SA level and supports fungal virulence [22]. Although we have shown the interaction of ZmCM2 with Vag2, this does not necessarily mean that the effect of Vag2 on virulence is via this interaction or this interaction alone. Several other putative interaction partners have been identified in the yeast two-hybrid screen (Table 1) that, if their interaction can be proven, could also contribute to the function of Vag2. Further experiments are needed to identify the exact function of Vag2 and confirm its proposed effect on the enzyme activity of ZmCM2.

## Figures and Tables

**Figure 1 jof-08-00498-f001:**
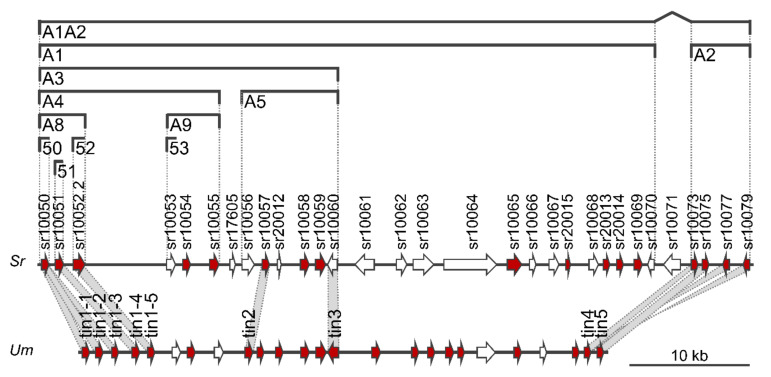
Map of divergence region 19-1 of *Sporisorium reilianum* including delineation of subcluster deletions and comparison to the gene cluster 19A of *Ustilago maydis*. Arrows indicate open reading frames. Genes predicted to encode secreted proteins (SignalP 6.0) are colored in red. Gene numbers from Genbank are indicated on top of each gene. Regions deleted in subcluster deletion strains are indicated by brackets. The name of the deletion is indicated below the bracket. Region 19-1 of *S. reilianum* (*Sr*, top) is compared to cluster 19A of *U. maydis* (*Um*, bottom [44]). Genes identified as virulence genes in *U. maydis* are named, and the homology of these genes to *S. reilianum* is indicated by grey shading.

**Figure 2 jof-08-00498-f002:**
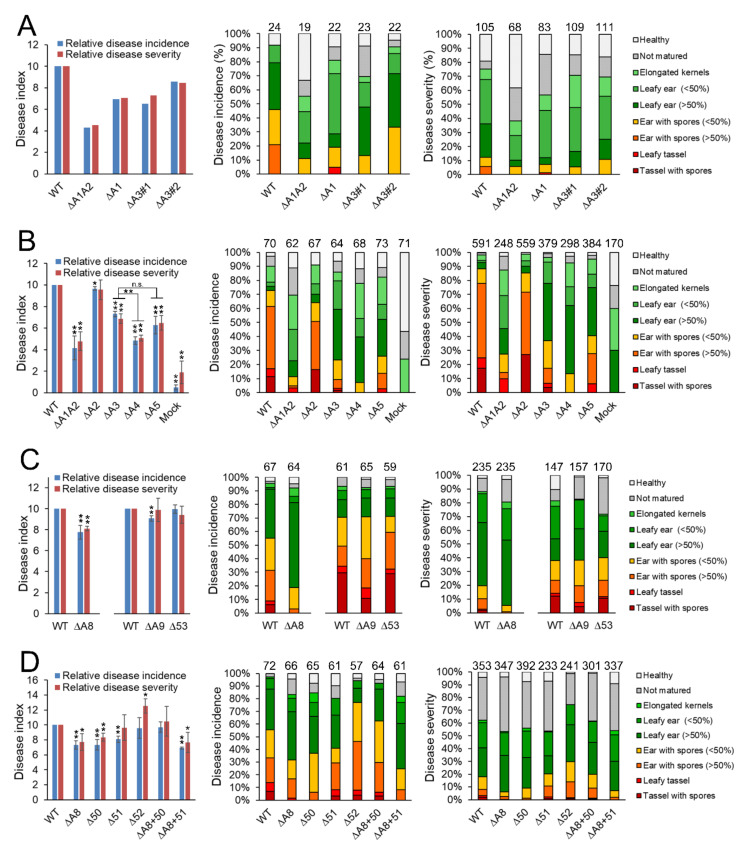
Dissection of virulence genes within divergence region 19-1 of *S. reilianum*. Virulence phenotype of subcluster gene deletion strains of *S. reilianum* on *Zea mays* cv. ‘Gaspe Flint’. Comparison of disease incidence and severity indexes (**left**), disease incidence distribution (**middle**), and disease severity distribution (**right**) are shown. Disease incidence gives the number of plants with the strongest displayed symptom (N, total number of evaluated plants), whereas disease severity gives the number of inflorescences displaying the strongest symptom (N, total number of evaluated inflorescences). N is indicated above the bar graph columns. The disease index indicates the average weighted strength of the displayed disease symptoms per total number of plants (incidence) or per total number of inflorescences (severity) relative to the respective values induced by wildtype (WT) infections, that were set to 10. Data are listed in Appendix A. Student’s *t*-test was used for statistical analysis. * *p* < 0.05; ** *p* < 0.01. (**B**–**D**) represent cumulative data of three independent biological replicates testing about 20 plants for each replicate. Error bars indicate SEM. Strains used are listed in Appendix A. Mock indicates that plants were inoculated with water. (**A**) Virulence of subcluster A3 gene deletion strains in comparison to subcluster A1 deletion strains; (**B**) Virulence of subcluster A4 and A5 gene deletion strains; (**C**) Virulence of subcluster A8 and A9 gene deletion strains; (**D**) Virulence of region A8 gene deletion strains. Strains are lacking all three genes of region A8 (ΔA8), only the first gene *sr10050/vag2* (Δ50), only the second gene *sr10051* (Δ51), or only the third gene *sr10052.2* (Δ52). ΔA8 complementation strains contain either the gene *sr10050/vag2* (ΔA8 + 50) or *sr10051* (ΔA8 + 51) in the ΔA8 region.

**Figure 3 jof-08-00498-f003:**
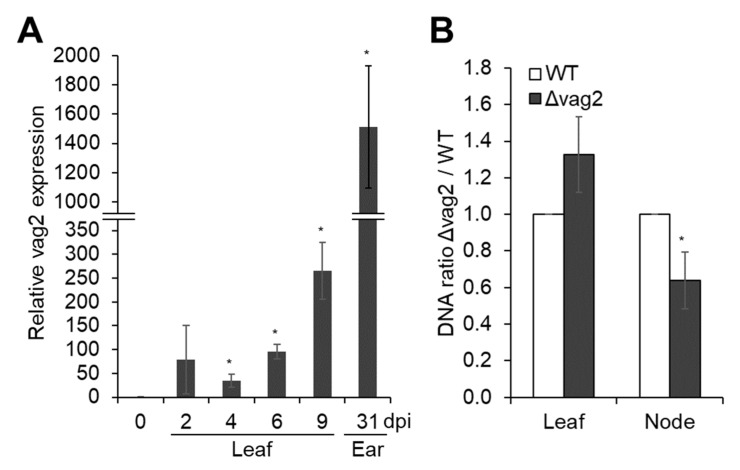
Quantification of *vag2* mRNA and fungal DNA in inoculated maize seedlings. (**A**) Relative expression of *vag2* (*sr10050*) in *S. reilianum*-inoculated *Z. mays* cv. ‘Gaspe Flint’ tissues. Samples were harvested from a mating mixture of sporidia grown in liquid culture (0 dpi), from inoculated maize leaves at 2, 4, 6, and 9 dpi, or from ears of inoculated plants at 31 dpi. Total RNA isolated from collected samples was used for quantitative RT-PCR. The expression of *vag2* in all samples was normalized to expression values of the *S. reilianum* glyceraldehyde 3-phosphate dehydrogenase (GAPDH) gene (*sr10940.2*) and is represented relative to the expression value of axenic culture. Error bars represent the SEM of three biological replicates. Each biological replicate is a pool of at least 10 tissue samples of independent plants. (**B**) Proliferation density of *Δ**vag2* deletion strains (Δvag2) in leaves and nodes of colonized *Z. mays* cv. ‘Gaspe Flint’ relative to wildtype (WT) strains. Total DNA was isolated from 3-cm pieces of inoculated leaves at 3 dpi and of nodes at the base of inoculated leaves at 14 dpi. The fungal GAPDH and maize actin genes were used to quantify relative fungal proliferation by quantitative PCR. The ratio of the fungal proliferation of *Δ**vag2* deletion strains relative to wildtype strains was calculated as a measure of proliferation density. Error bars represent the SEM of three biological replicates. Each biological replicate is a pool of 10 samples from different plants. Data were analyzed with Student’s *t*-test (* *p* < 0.05).

**Figure 4 jof-08-00498-f004:**
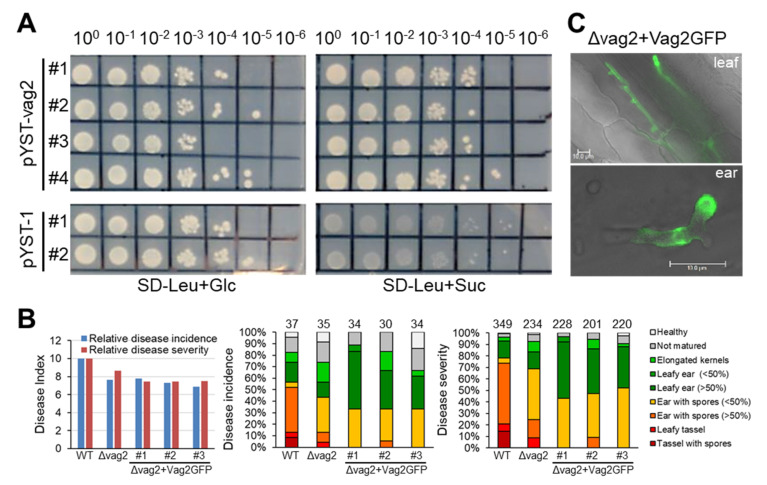
Vag2 contains a secretion signal peptide, but a Vag2-GFP Fusion is not functional. (**A**) Yeast secretion trap assay confirms the predicted secretion signal peptide of Vag2. *Saccharomyces cerevisiae* SEY6210 was used for transformation with the empty vector containing the *SUC2* gene and lacking the N-terminal sequences encoding the secretion signal peptide (pYST-1), or with the pYST-1 vector containing the Vag2 open reading frame in fusion to the *SUC2* gene of pYST-1 (pYST-vag2). Serial dilutions of different transformants were plated on SD minimal medium lacking leucine but containing glucose (SD-Leu + Glc) as growth control, and on SD minimal medium lacking leucine but containing sucrose (SD-Leu + Suc) as a test. Only strains containing *vag2* were able to use sucrose as a carbon source. (**B**) A Vag2-GFP fusion does not complement the lack of Vag2 during maize colonization by *S. reilianum*. Seedlings of *Z. mays* cv. ‘Gaspe Flint’ was used for inoculation with combinations of mating-compatible wild-type strains (WT), strains lacking *vag2* (Δvag2), or strains lacking *vag2* and containing a *vag2-GFP* fusion gene at the native locus of *vag2* (Δvag2+Vag2GFP). Three different combinations of strains were used (Appendix A). Comparison of disease incidence and severity indexes (**left**), disease incidence distribution (**middle**), and disease severity distribution (**right**) are shown, see legend in Figure 2 for more explanations. (**C**) Fluorescence microscopic analysis of fungal hyphae expressing Vag2-GFP instead of Vag2 (Δvag2+Vag2GFP). Pictures represent merged bight field and GFP images of hyphae of *S. reilianum* colonizing leaves at 3 dpi (**top**) or ears at 31 dpi (**bottom**). Size bars: 10 µm.

**Figure 5 jof-08-00498-f005:**
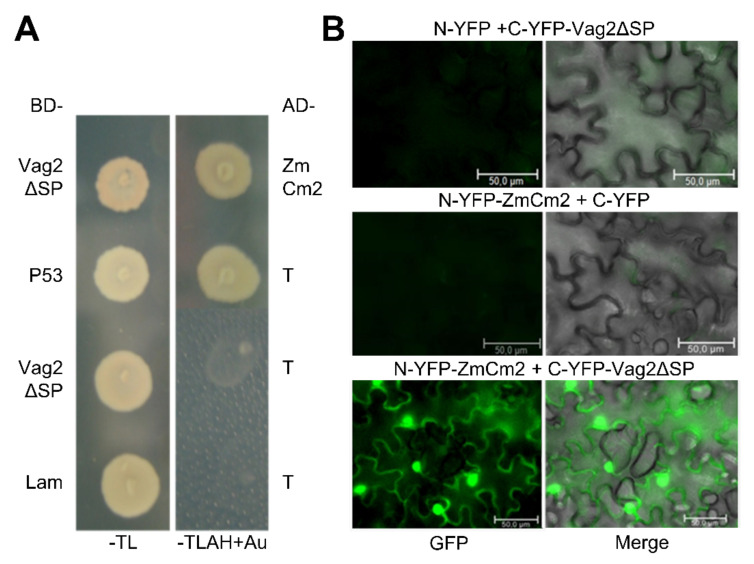
Vag2 interacts with ZmCM2. (**A**) Vag2 lacking its signal peptide was cloned in fusion with the GAL4 binding domain (BD-Vag2ΔSP) and introduced into the Y2HGold strain. One of the retrieved plasmids containing maize chorismate mutase-derived sequences in fusion to the GAL4 activation domain (AD-ZmCM2) was introduced into Y187. Both strains were mated, and diploids were selected on SD minimal media lacking tryptophan and leucine (-TL). Reconstitution of the GAL4 transcription factor by the interaction of Vag2ΔSP with ZmCM2 was verified by checking the expression of the GAL4-promoter controlled genes ADE2 and HIS3 as well as a gene allowing resistance to the drug aureobasidine A on SD medium lacking tryptophan, leucine, adenine and histidine, and containing aureobasidine A (-TLAH + Au). As a positive control, strains expressing tumor suppressor p53 (AD-P53) and the strong interaction partner SV40 T-antigen (BD-T), were used. As negative controls, strains were expressing Vag2 (BD-Vag2ΔSP) or Lamin (AD-Lam) and T-antigen (BD-T). (**B**) Bimolecular fluorescence complementation confirms in-planta interaction between Vag2 and ZmCM2. The N-terminal part of YFP was expressed alone (N-YFP) or as a fusion with ZmCM2 (N-YFP-ZmCM2), whereas the C-terminal part of YFP was expressed alone (C-YFP) or as a fusion with Vag2 lacking its signal peptide (C-YFP-Vag2ΔSP) after infiltration of *Nicotiana benthamiana* leaves with *Agrobacterium tumefaciens* cells delivering the indicated constructs. At 3 days after infiltration, the leaves were analyzed by fluorescence microscopy. YFP fluorescence was only seen when both fusion proteins were co-expressed. Shown are representative cells under conditions enabling detection of GFP fluorescence (**left**), and a merge of the GFP fluorescence and brightfield pictures (**right**). Size bars: 50 µm.

**Figure 6 jof-08-00498-f006:**
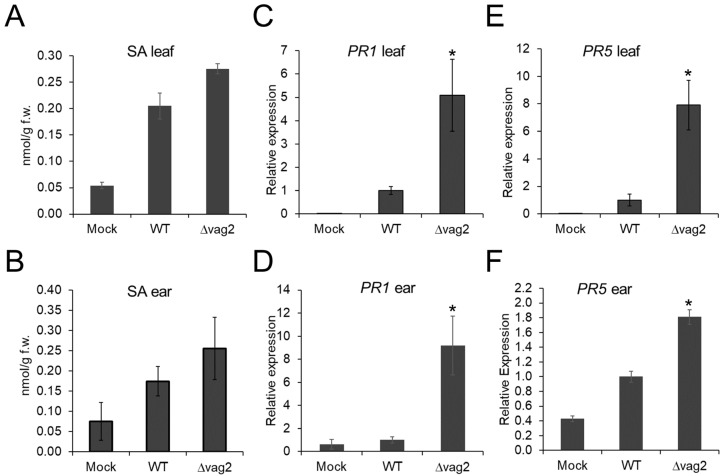
SA and SA-induced defense genes are upregulated in maize colonized by *S. reilianum* lacking *vag2*. (**A**,**B**) SA levels were quantified in maize tissues inoculated with *S. reilianum* wildtype (WT) and *Δ**vag2* deletion strains (Δvag2), or with water (Mock). Samples were collected from maize leaves below the inoculation site at 6 dpi (**A**), or of maize ears collected at 31 dpi (**B**). (**C**–**F**) Relative expression of the SA-induced defense genes *PR1* (AC205274.3_FG001) and *PR5* (GRMZM2G402631) was measured by real-time PCR. Samples were collected from inoculated leaves at 6 dpi (**C**,**E**) and from ears at 31 dpi (**D**,**F**) and used for extraction of total RNA. RNA samples were subjected to qRT-PCR with *PR1* and *PR5* gene-specific primers (Appendix A). Expression levels of *PR1* (**C**,**D**) and *PR5* (**E**,**F**) were normalized to the maize actin reference gene (GRMZM2G126010), and expression of WT was set to 1. For each biological replicate, leaves or ears of 10 different plants were collected. Error bars indicate SEM of three biological replicates, and asterisks indicate a significant difference (Student’s *t-*test, * *p* < 0.05) of mutant relative to wildtype-inoculations.

**Figure 7 jof-08-00498-f007:**
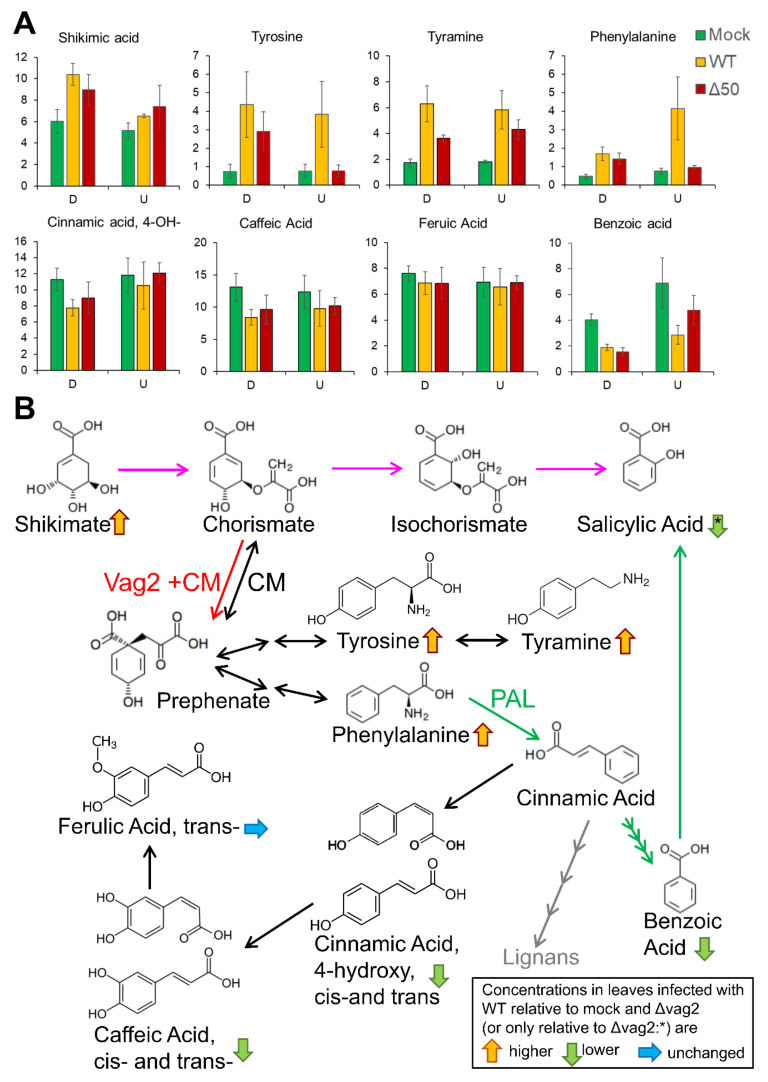
Model of the possible function of Vag2. (**A**) Visualization of metabolite quantification based on GC-EI/TOF-MS analysis of total metabolome of control-inoculated maize seedlings (Mock) or maize seedlings inoculated with *S. reilianum* wildtype strains (WT) or strains lacking *vag2* (Δ50). Sections of inoculated leaves were collected 3 days post-inoculation and consisted of the colonized part (D) below the inoculation marks and the non-colonized part of the leaf above the inoculation wounds (U). Data show averages of seven to eight individual leaf samples. Error bars indicate standard error. The values depicted on the Y-axis are arbitrary abundance units used to determine metabolite amounts relative to a spike of ^13^C_6_-sorbitol. The *cis*- and *trans*- forms of 4-hydroxy-cinnamic acid and of caffeic acid were measured individually and summed before the calculation of error bars. Statistical analysis indicated no significant differences between wildtype- and *Δvag2*-inoculated samples (Student’s *t*-test, *p* < 0.05). (**B**) Schematic depiction of the two main pathways of SA generation in plants and how Vag2 might influence metabolite flow in modulating the activity of the maize chorismate mutase ZmCM2. The main route of SA generation in plant-microbe interactions may be from shikimate via chorismate and isochorismate (pink arrows). The second possibility is from phenylalanine via cinnamic acid and benzoic acid (green arrows). The chorismate mutase (CM) catalyzes the reversible interconversion of chorismate and prephenate, a precursor of tyrosine, tyramine, and phenylalanine. Phenylalanine is a substrate of the phenylalanine-ammonia-lyase (PAL), the key enzyme for SA generation via the second route. Metabolite analysis depicted in (**A**) indicates that when maize seedlings were infected with *S. reilianum* wild type, the levels of tyrosine, tyramine, phenylalanine, and shikimate were elevated relative to that of control plants and relative to plants inoculated with *vag2* deletion strains, whereas the levels of benzoate, 4-hydroxy-cinnamate and caffeate were slightly lower. SA levels were shown to be increased in *Δvag2*-inoculated samples relative to wildtype-inoculated samples (Figure 6A,B). Vag2 interaction with ZmCM2 might make the ZmCM2-catalysed reaction unidirectional (red arrow), which would result in the redirection of the metabolite flow into the generation of tyrosine, tyramine, and phenylalanine, thus lowering the SA levels by lowering the SA precursor concentration.

**Table 1 jof-08-00498-t001:** List of Vag2 interaction partners identified by yeast two-hybrid screening of a cDNA library generated from *S. reilianum*-colonized maize tissues.

Interaction Partner	Frequency	Gene ID	Description
Metabolism			
IP1	24	GRMZM2G179454	Zea mays chorismate mutase (ZmCm2)
IP2	11	GRMZM2G027663	Putative ThiC superfamily protein
IP3	10	GRMZM2G006329	Zea mays Enzyme: pleckstrin homology (PH) domain-containing protein
IP4	6	GRMZM2G135588	Putative citrate synthase family protein
IP5	4	GRMZM2G081585	Chloroplastic iron-superoxide dismutase (sodB)
IP6	3	AC208571.4_FG001	Hypothetical protein containing a haloacid dehalogenase-like hydrolase family domain / NHL repeat domain
IP7	3	GRMZM2G300862	Aspartate kinase
IP8	3	GRMZM2G014788	Unknown protein containing Carboxypeptidase regulatory-like domain
IP9	2	GRMZM2G135588	Aspartate kinase homoserine dehydrogenase 2 (akh2)
IP10	2	GRMZM2G049538	Terpene synthase1
IP11	1	GRMZM2G151934	Zea mays protein DA1-related 2-like
IP12	1	GRMZM2G043198	Pyruvate dehydrogenase 2 (pdh2)
IP13	1	GRMZM2G121612	Starch synthase
IP14	1	GRMZM2G118806	Uncharacterized protein with proteolysis and peptidase activity
IP15	1	GRMZM2G088689	2-Oxoisovalerate dehydrogenase (acylating)
IP16	1	GRMZM2G448142	Putative NADH-quinone oxidoreductase subunit K
Transcription/DNA binding			
IP17	7	GRMZM2G100246	Unknown protein containing NOT2,3,5 domain
IP18	4	GRMZM2G440943	Helicase/SANT-associated, DNA binding protein
IP19	2	GRMZM2G351304	Uncharacterized protein containing chromosome segregation protein SMC domain
IP20	2	AC225308.2_FG005	Putative homeodomain-like transcription factor superfamily protein
IP21	1	GRMZM2G133016	MYB DNA-binding domain superfamily protein
IP22	1	GRMZM5G876621	Zea mays putative RING zinc finger domain superfamily protein
IP23	1	GRMZM2G377369	Uncharacterized protein containing DNA binding site
IP24	1	AC226373.2	Zink finger C-x8-C-x5-C-x3-H type family protein
IP25	1	GRMZM2G340749	General negative regulator of transcription
Protein processes			
IP26	5	GRMZM2G027282	Proteasome 26S subunit 6A (RPT5a)
IP27	3	GRMZM2G168119	Putative HSP20-like chaperone domain family protein
IP28	2	GRMZM2G134980	Putative dnaJ chaperone family protein
IP29	2	GRMZM2G006781	Conserved oligomeric Golgi complex subunit 8
IP30	1	GRMZM2G551402	Unknown protein containing a ubiquitin carboxyl-terminal hydrolase domain
IP31	1	GRMZM2G012631	HSP protein (HSP90-2)
IP32	1	GRMZM2G137495	DnaJ domain or J-domain. DnaJ/Hsp40 (heat shock protein 40)
IP33	1	GRMZM2G162968	Chaperone protein ClpB2
IP34	1	GRMZM2G154312	Co-chaperone protein SBA1
Signaling			
IP35	4	GRMZM2G038982	Uncharacterized protein containing a STKc_MAP3K-like domain
IP36	3	GRMZM2G126946	Zea mays putative calcium-dependent lipid-binding (CaLB domain) family protein
IP37	1	GRMZM2G152877	Uncharacterized protein containing F-box-like and Cupin-like domain domain
IP38	1	GRMZM2G326472	Uncharacterized protein containing a STKc_MAP3K-like domain
Nuclear processes			
IP39	2	GRMZM2G159028	RNA binding protein
IP40	1	GRMZM2G111014	Unknown protein containing DNA gyrase subunit
IP41	1	GRMZM2G588223	Hypothetical protein ZEAMMB73 containing double-stranded RNA binding motif
IP42	1	GRMZM2G030128	DNA repair-recombination protein (rad50)
IP43	1	AC205703.4_FG010	Hypothetical protein ZEAMMB73_142911/ATPase involved in DNA replication, recombination, and repair

## Data Availability

Not applicable.

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
