# Peer review of "The Sporisorium reilianum Effector Vag2 Promotes Head Smut Disease via Suppression of Plant Defense Responses"

_jof, 2022, doi:10.3390/jof8050498_

Round 1
Reviewer 1 Report
The experiments are well designed, and the manuscript is well written. I recommend the manuscript to be accepted with minor revisions.
- Based on Figure 5, ZmCM2 is distributed across the whole cell, from the plasma membrane, cytoplasm to nucleus. However, in the Discussion section (pages 742-743), ZmCM2 is located in cytoplasm, please give an explanation to this inconsistency.
- It is clear that in Figure 2D, ΔA8+50 showed the same disease severity with the wild-type, suggesting 50 could complement the deletion of A8 to confer full pathogenic effect on maize. Whereas, in Figure 4B, 50GFP failed to complement ​Δ50, why? Did the authors test the disease severity of Δ50+50?
- In the process of interaction, the host gene is high likely to be induced to counter-attack the pathogen invasion. I suggest the authors investigated the mRNA or protein levels of ZmCM2 in response to Vag2 infection.
Reviewer 2 Report
This manuscript "The Sporisorium reilianum effector Vag2 promotes head smut disease via suppression of plant defense responses" presented by Zhao et al. is very well-prepared, especially it is written in professional English, which makes understanding easier for readers.
Generraly speaking, the manuscript is technically sound, and the data supports the conclusions. The statistical analysis been performed appropriately and rigorously. Besides, the strengths of the study is good, so as the validity of the methods, results, and data interpretation.
From all the above, the present version well meets the criteria of Journal of Fungi and can be accepted for publication in its present form.
Reviewer 3 Report
The manuscript authored by Zhan et al., identified an S. reilianum effector vag2, which affected the pathogen virulence. Further they identified the cytoplasmic maize chorismate mutase ZmCM2 as an interaction partner of Vag2 and confirmed their interaction by bimolecular fluorescence complementation. They performed gene expression and metabolite analyses, and concluded that Vag2 suppresses plant defense through interfering with SA pathway by interacting with maize ZmCM2.
This is a nice study overall. The experiments were well designed, results and data were clearly explained. The conclusions were drawn reasonably. I do not have major concerns for the manuscript.
A few minor suggestions.
Fig 3B: The figure should be made in leaf (wild type/mutant) and node (wild type/mutant). The presented figure looks weird. You should be able to switch column if the figure was made in Excel.
Fig 4B and C: It should be easier to follow using ∆vag than ∆50. How do you explain that the complementation of vag did not work?
Fig 6 A, please label significant (* is missing?) if the differences are significant as described in line 563-564.
Line 65 and other lines: suggest change effector proteins to effectors. You use effectors in the following paragraphs, to be consistent, use effectors.
Line 333: delete each of (duplicated), add in or for
Line 708: the highest
Reviewer 4 Report
Excellent and interesting work.
- The research carried out an analysis of effectors produced by Sporisorium reilianum through the comparación with Ustilago maydis
- and also how these contributed in the suppression of plant defense and virulence. The authors identified an effector called “Vag2” with an evident effect during the interaction. The results support the role of “Vag2” in the virulence and suppression of SA pathway.
- The new finding is related with the effect displayed by Vag2 during the interaction and how this effector suppress the plant defense mediated by salicylic acid pathway.
- The manuscript is well-written and the experiments were well-envisage.
The different experiments follow a logical way.
- The results and new findings are clear and allow the development of new strategies to control the pathogen taking account the role of this specific effector in the virulence.
